# Marked isotopic variability within and between the Amazon River and marine dissolved black carbon pools

Alysha I. Coppola [1], Michael Seidel [2], Nicholas D. Ward [3,4], Daniel Viviroli [1], Gabriela S. Nascimento[1,5], Negar Haghipour[5,6], Brandi N. Revels[5], Samuel Abiven [1], Matthew W. Jones[7], Jeffrey E. Richey [4], Timothy I. Eglinton [5], Thorsten Dittmar [2,8] & Michael W.I. Schmidt [1]

Riverine dissolved organic carbon (DOC) contains charcoal byproducts, termed black carbon (BC). To determine the significance of BC as a sink of atmospheric $CO_2$ and reconcile budgets, the sources and fate of this large, slow-cycling and elusive carbon pool must be constrained. The Amazon River is a significant part of global BC cycling because it exports an order of magnitude more DOC, and thus dissolved BC (DBC), than any other river. We report spatially resolved DBC quantity and radiocarbon ($\Delta^{14}C$) measurements, paired with molecular-level characterization of dissolved organic matter from the Amazon River and tributaries during low discharge. The proportion of BC-like polycyclic aromatic structures decreases downstream, but marked spatial variability in abundance and $\Delta^{14}C$ values of DBC molecular markers imply dynamic sources and cycling in a manner that is incongruent with bulk DOC. We estimate a flux from the Amazon River of 1.9–2.7 Tg DBC yr$^{-1}$ that is composed of predominately young DBC, suggesting that loss processes of modern DBC are important.

[1] Department of Geography, University of Zurich, Winterthurerstrasse 190, 8057 Zürich, Switzerland. [2] Research Group for Marine Geochemistry, Institute for Chemistry and Biology of the Marine Environment (ICBM), University of Oldenburg, D-26129 Oldenburg, Germany. [3] Marine Sciences Laboratory, Pacific Northwest National Laboratory, 1529 West Sequim Bay Road, Sequim, WA 98382, USA. [4] School of Oceanography, University of Washington, Box 355351, Seattle, WA 98195, USA. [5] Geological Institute, Department of Earth Sciences, ETH Zürich, Sonneggstrasse 5, 8092 Zürich, Switzerland. [6] Laboratory of Ion Beam Physics, ETH Zürich, Otto-Stern-Weg 5, 8093 Zürich, Switzerland. [7] Tyndall Centre for Climate Change Research, School of Environmental Sciences, University of East Anglia, Norwich NR4 7TJ, UK. [8] Helmholtz Institute for Functional Marine Biodiversity at the University of Oldenburg (HIFMB), Ammerländer Heerstraße 231, 26129 Oldenburg, Germany. Correspondence and requests for materials should be addressed to A.I.C. (email: Alysha.coppola@geo.uzh.ch)

Biomass burning and fossil fuel combustion release vast amounts of carbon into the atmosphere, causing large changes in Earth's climate[1,2]. Up to 27% and 0.2% of carbon from the incomplete combustion of biomass and fossil fuel, respectively, is retained as condensed forms of carbon (called pyrogenic or black carbon, BC, ranging from charcoal to soot) rather than emitted as greenhouse gases[3]. In addition to impacting radiative budgets, BC also influences biogeochemical processes because it is a very large and refractory component of the global carbon cycle[4,5]. For example, particulate BC (PBC) acts as a biospheric carbon sink[3,6,7] by removing carbon from faster atmosphere–biosphere processes and sequestering this carbon to sedimentary reservoirs.

To predict how the carbon cycle may respond to climate change, we need to determine the origin, dynamics, and fate of this abundant and slowly cycling component in the carbon cycle. A large portion of BC is exported to the ocean by rivers as dissolved BC (DBC) (27 Tg year$^{-1}$)[8], thereby connecting marine and terrestrial carbon cycles[8,9]. Atmospheric deposition is another major pathway in which BC reaches rivers (after mobilization from the landscape)[10] and also the ocean (1.8–10 Tg year$^{-1}$)[11,12]. Uncertainties in regional and global-scale BC budgets persist due to poor constraints on its fluvial dynamics and export. Current estimates suggest that the input by rivers alone to the ocean is sufficient to sustain the turnover of the entire oceanic BC pool in just 500 $^{14}$C years given current known losses of BC, yet measured $^{14}$C ages of BC in the deep sea are 40 times greater (>20,000 $^{14}$C years)[13–15]. Our understanding of the role of BC in the regional and global-scale carbon cycle remains inadequate, due in large part to poor constraints on the processing, quality, and fate of DBC during river export to the ocean[16].

Rivers are the primary link by which BC is transferred laterally from terrestrial pools to the oceans[9,16]. Currently, large gaps exist with respect to the processes, pathways, and timescales over which DBC is mobilized and transported from land to ocean[16–19], leading to challenges in reconciling BC cycling on terrestrial landscapes (modern to 1000 years mean residence time) with its longevity in the deep oceans (>20,000 $^{14}$C years)[14]. The annual export of DBC is large (27 Tg/year), and represents a significant fraction (i.e., ~10%) of overall riverine dissolved organic carbon (DOC)[9]. Surprisingly, riverine DOC and DBC concentrations appear to be coupled[9,17,18,20] regardless of fire history in upstream catchments, and despite substantial lags between the production of charcoal BC and river DBC export[21–23]. For example, in the Paraíba do Sul River in Brazil, DBC is continuously mobilized irrespective of fire history, with annual export far exceeding contemporary BC production rates[21]. DBC is partially removed during mobilization from soils during the wet season[22], implying storage of BC in intermediate reservoirs (such as soils) prior to release to the river network[24]. BC has residence times of centuries in soils[25], while turnover times are on the order of millennia in the deep ocean[13,14]. These contrasting turnover times suggest that aging may occur along the land–river–ocean transport continuum. Alternatively, more reactive DBC pools could be selectively removed during storage and transport, resulting in an apparent increase in age of residual DBC. For example, low-temperature combustion products derived from modern wildfires are re-mineralized during transport in Arctic river systems on the timescale of 20–40 days[26]. However, analyzing the DBC radiocarbon composition ($\Delta^{14}$C) is needed to understand how these remineralization processes influence the age distribution and recalcitrance as DBC is transported from river to ocean reservoirs. On a global scale, PBC in rivers is refractory, but $\Delta^{14}$C measurements on different river systems suggest that the extent of storage within river catchments varies widely for PBC pools[8]. Information on the radiocarbon composition of riverine DBC is currently limited to a single study[17].

The Amazon River accounts for one-fifth of global freshwater discharge to the ocean[27], and is the largest single source of terrestrial organic matter to the ocean (with an average annual DOC export of 22–27 Tg)[28,29]. Thus, the Amazon River is a crucial system in which to understand DBC cycling and transport and to develop constraints on global BC dynamics. The Amazon Basin transitions between pristine forest and urban influenced aerosol polluted plumes due to rapid developments in energy, agriculture expansion, and deforestation[30]. Here, we collected dissolved organic matter (DOM) and DBC samples for molecular characterization and radiocarbon analysis in four tributaries (Negro, Madeira, Trombetas, and Tapajós Rivers) and the Amazon River mainstem in November 2015 during one of driest seasons following a strong El Niño (Methods Sampling and Site Locations, Supplementary Fig. 1). This period represents the low flow of water during which the floodplain complexity and connectivity is limited, as we sampled only permanent waters[31]. Although DOC export is greatest during intervals of peak discharge[32], we selected this low flow period for analysis to reduce complexity in watershed DBC sources associated with floodplain dynamics. We measured the concentration and radiocarbon content ($\Delta^{14}$C values) of DBC in solid phase-extracted (SPE) DOC using molecular proxies (benzene polycarboxylic acids, BPCAs)[33] released through chemical oxidation from the polycyclic condensed aromatic structure of BC. We pair these quantitative measurements of DBC molecular markers with DOM molecular characterization using ultrahigh-resolution mass spectrometry (Fourier transform ion cyclotron resonance mass spectrometry; FT-ICR-MS)[34–36]. DBC $\Delta^{14}$C measurements reveal that DBC within the Amazon River and its tributaries generally are modern in age, but considerable spatial heterogeneity exists. Notably, there are very low DBC $\Delta^{14}$C values at Trombetas-Oriximina, Amazonas-Santarem, Óbidos, and near Manaus. However, further downstream, the Amazon River exports modern DBC to the ocean.

## Results

**Spatial decoupling of DBC and DOC concentrations and trends.** We observe incongruent dynamics of DBC and DOC concentrations. DOC concentrations along the mainstem range from 3.3 ± 0.3 mg L$^{-1}$ upstream to 2.5 ± 0.2 mg L$^{-1}$ downstream at Manacapuru (station 11) to Almeirim (station 1), respectively (Supplementary Fig. 2). Tributary concentrations range from 1.6 ± 0.5 to 6.8 ± 0.5 mg L$^{-1}$ in the Tapajós River (station 2) to Negro River (station 12), respectively. The DOC flux along the Amazon River mainstem increases from 183 kg C s$^{-1}$ at Manacapuru (station 11) to 229 kg C s$^{-1}$ at Almeirim (station 1) (Supplementary Table 1, Supplementary Fig. 2). With the exception of an Amazon River mainstem sample near Parintins that has a significantly higher DBC concentration (802 ± 160 μg L$^{-1}$, station 6), DBC concentrations range from 103 ± 21 μg L$^{-1}$ in the Tapajós River tributary (station 2), to 181 ± 36 μg L$^{-1}$ at Almeirim (station 1) to 495 ± 90 μg L$^{-1}$ at Santarém (station 3) (Fig. 1). The weighted average DBC concentration is 264 ± 20 μg L$^{-1}$ (22 ± 4 μM) along the mainstem ($n = 8$). There is also a general increase in DBC fluxes downstream from Manacapuru (station 11) to Almeirim (station 1) from 13.3 to 16.6 kg s$^{-1}$, respectively (Supplementary Table 2, Supplementary Fig. 2). This lack of covariation between DBC and DOC concentrations along the river ($p = 0.59$) during low flow contrasts with observations from other rivers globally[9,20,37]. Although documented in other river catchments[35,38], this decoupling between DBC and DOC concentrations may be a consequence of the extreme drought conditions throughout the

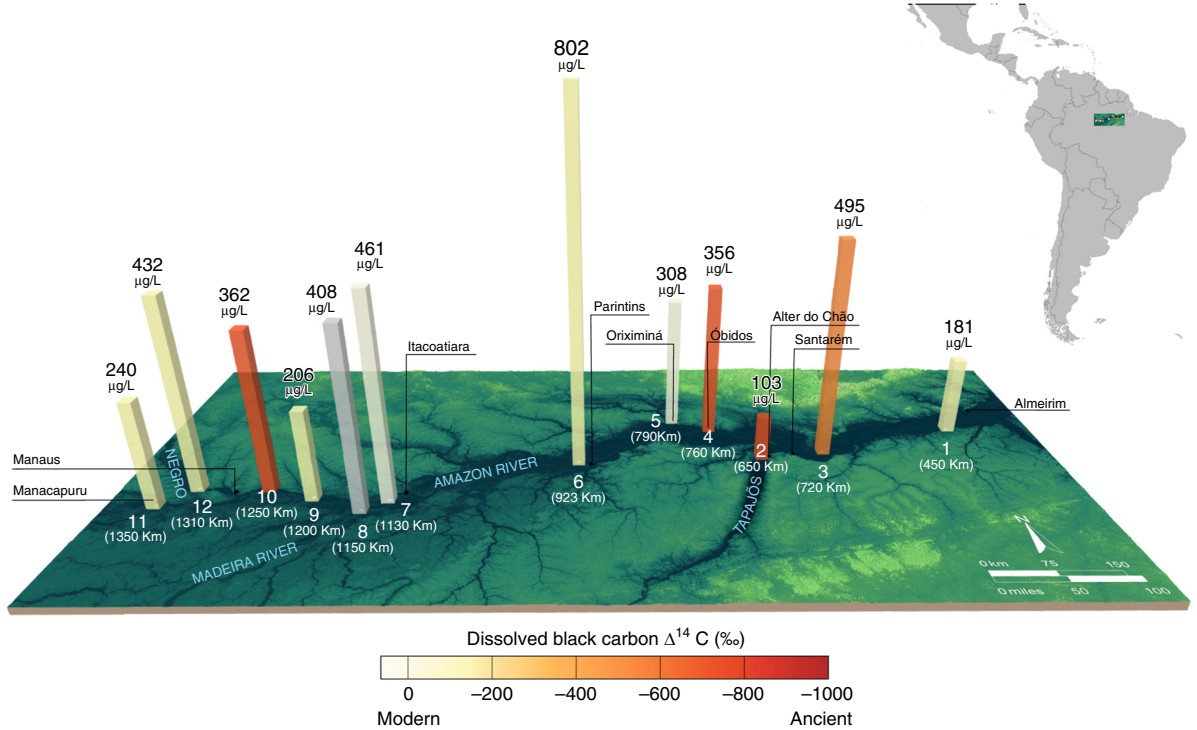

**Fig. 1** Dissolved black carbon concentration and radiocarbon values. Sample locations plotted as distance to the river mouth (in km, in white text) with the river (in blue text) and nearest city (black text). Dissolved black carbon (DBC) concentrations are shown by the height of the columns in μg DBC L$^{-1}$. The shades of yellow to red of columns represents modern to ancient DBC $\Delta^{14}$C values. Station locations (in white) are labeled 1–12 according to Supplementary Fig. 1. The gray column at station 8 indicates no DBC $\Delta^{14}$C data. Error bars (s.d) are given in Supplementary Table 1

Amazon basin in 2015[39,40], which may have affected the dynamics of DOC in this catchment. The lack of correlation between DBC and DOC concentrations in tropical rivers ($p < 0.6023$) may result from the divergent effects of soil properties, temperature, rainfall, and aerosol deposition on bulk DOC and its DBC mobilization from catchments[38,41].

**Isotopic heterogeneity of DBC.** SPE-DOC shows $\Delta^{14}$C values ($-10 \pm 24$‰ to $+55 \pm 30$‰) consistent with modern biospheric inputs at all stations (Supplementary Fig. 3). The latter coupled with corresponding stable carbon isotopic compositions ($\delta^{13}$C values $-28.5 \pm 0.8$‰ to $-31.5 \pm 0.1$‰) (Supplementary Table 1) suggests contemporary lowland C3 plants as the dominant source of DOC[42,43]. Also, DBC $\Delta^{14}$C values are mostly consistent with a modern river source[13] (weighted average, $-46 \pm 15$‰, $n = 7$). However, DBC $\Delta^{14}$C values are markedly lower at four sites (station 10, Amazonas-20 km downstream of Manaus $-720 \pm 8$‰; station 4, Amazonas-Óbidos, $-658 \pm 7$‰; station 3, Amazonas-Santarém $-431 \pm 14$‰; station 2, Tapajós-Alter do Chão, $-771 \pm 16$‰ (Fig. 1, Supplementary Table 2)). The very low DBC $\Delta^{14}$C values suggest that DBC is not stored in short-term intermediate reservoirs prior to export. The DBC $\Delta^{14}$C values are not correlated to DOC flux ($p = 0.260$), discharge ($p = 0.36$), or other catchment-specific parameters, such as land cover ($p = 0.385$ for wetlands, $p = 0.094$ for croplands, $p = 0.516$ for natural vegetation) burned ($p = 0.283$) or urbanized area ($p = 0.985$) (Supplementary Fig. 4, Supplementary Table 3). The low DBC $\Delta^{14}$C values are also not reflected in abundances of DBC molecular markers (Supplementary Fig. 5). There are several potential explanations for the presence of these four strongly $^{14}$C-depleted riverine DBC values. For example, there is an anthropogenic aerosol plume 8–70 km downwind of Manaus from

regional urbanization[44] that may contribute to the low $\Delta^{14}$C DBC values we observe at station 10 (20 km downwind)[45]. A second possible explanation for low DBC $\Delta^{14}$C values in mainstem sites just downstream of tributaries (e.g., Santarém) is that the collected water represented an impartial mixture of tributary and mainstem water considering that tributaries in the Amazon have been shown to be poorly mixed up to 100 km downstream of their confluence[46]. Additionally, although past measurements show that suspended POC in the lowland rivers studied here have modern radiocarbon signatures similar to DOC[43], PBC in the Amazon River has a $\Delta^{14}$C value of $-386 \pm 43$‰ (3900 $\pm$ 770 $^{14}$C years)[8]. This suggests that the old DBC values we observe here may have some contribution derived from sedimentary material desorbed into the dissolved phase. Overall, the sharp isotopic variability within the DBC brings up more questions than answers. However, in absence of further data we cannot reach a conclusion (see Supplementary Discussion for further information). Future field, laboratory and modeling studies to map urban and biomass burning emissions at different locations in the Amazon[47] paired directly with $\Delta^{14}$C aerosol BC, river PBC, sediments and molecular composition directly need to be conducted to fully determine the primary driver of the observed low DBC $\Delta^{14}$C values at these four sites to test these open hypothesizes.

**Compositional variability within DBC.** Of the ~12,000 molecular formulae detected in SPE-DOM using FT-ICR-MS, we approximate ~10% (1272 molecular formulae) contain a polycyclic aromatic (PCA) signature (aromaticity index, $AI_{mod} \geq 0.67$, $C \geq 15$), consistent with DBC-like sources. These PCA molecular formulae are strongly correlated to the DOM-normalized concentrations of B6CA marker compounds ($p \leq 0.001$, $r = 0.88$),

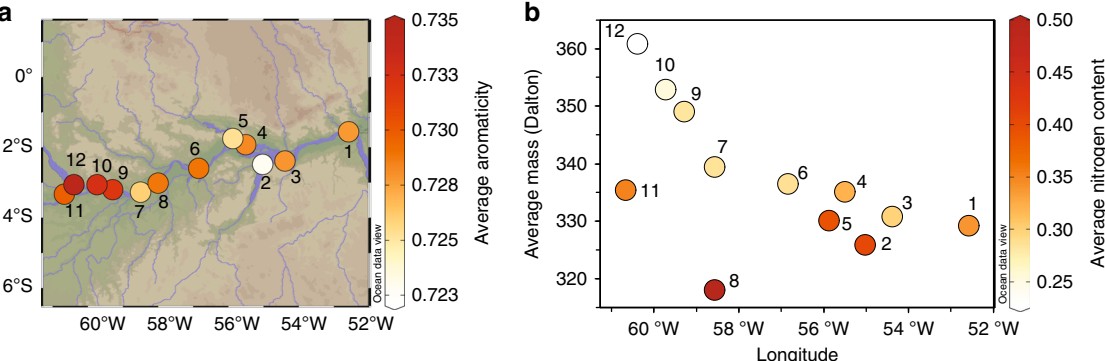

**Fig. 2** Polycyclic aromaticity and nitrogen content along the Amazon River. Molecular parameters of the 1272 detected polycyclic aromatic molecular formulae with aromaticity index ≥0.67 as analyzed by Fourier transform ion cyclotron resonance mass spectrometry (FT-ICR-MS). By definition, polycyclic aromatics include thermogenic DBC molecular formulae. Numbers on the plot correspond to station locations in Supplementary Table 1. **a** Plot of the intensity weighted-average aromaticity per samples of polycyclic aromatic molecular formulae along the sampling transect, given by shades of red. The aromaticity decreased relatively downstream. **b** The intensity weighted-average molecular mass (in Daltons) of polycyclic aromatic molecular formulae over the sampling transect with number of polycyclic aromatic molecular formulae with color-coded intensity weighted-average of nitrogen content of the polycyclic aromatic molecular formulae along longitude. The number of polycyclic aromatic molecular formulae decreased downstream while the relative content of nitrogen per polycyclic aromatic molecular formula increased

indicating a common origin from highly condensed compounds of pyrogenic and combustion sources. The weighted-average aromaticity of PCA molecular formulae decreases downstream (Fig. 2) concomitant with a decrease in more condensed BC structures given by higher B6CA marker concentrations and B6CA/B5CA ratios (Supplementary Fig. 3, Supplementary Note 1), suggesting downstream loss of more condensed BC structures by photooxidation[48–50]. The shift to smaller and less polycondensed DBC structures (Supplementary Fig. 6) may be a consequence of preferential removal of polycondensed PCA compounds due to photo-degradation[48] or sorptive interaction processes along the river continuum.

The relative increase of the nitrogen content in the poly-condensed aromatic molecular formulae (referred to as dissolved black nitrogen), may also suggest increased fossil fuel inputs[51]. The contribution of aerosol fossil fuel inputs that likely caused drastic changes in DBC Δ[14]C values along the river continuum is not reflected in the DOM molecular formulae. This suggests either similarities in PCA molecular formulae composition despite contrasting isotopic values and sources (i.e. fossil fuel vs. soils) or that the DBC changes are not detectable in bulk DOM molecular formulae[19]. The latter may be a consequence of restrictions in the analytical window of FT-ICR-MS due to differences in ionization efficiency of different DBC compounds derived from soot and charcoal inputs[19].

## Discussion

Based on the average annual Amazon River DOC export (22–27 Tg) and a range of DBC/DOC% values at low and high flow (7.2 ± 0.5%, at Almeirim, site 1; DBC/DOC% of 9.9 ± 1.0% at Macapa[52], respectively), we estimate an annual flux of DBC of 1.9–2.7 Tg from the Amazon River basin to the Atlantic Ocean (Methods, Supplementary Table 2). Based on the current estimate of 27 Tg year[−1] for global riverine DBC export[9], this corresponds to 7–10% of the global fluvial DBC flux to the oceans. This represents a lower estimate of DBC flux, based on a dry season sampling period from a station 450 km (Amazonas-Almeirim) upstream from the mouth of the Amazon, as compared to ~1175 km upstream peak discharge sampling location used in the global study[9] (as a composite sample of 25% Rio Negro and 75% Rio Solimões, at 3.13°S, 59.92°W, 400 µg L[−1], 9.5% DBC/DOC%).

We also report significantly lower DOC concentrations (2.5 ± 0.2 mg L[−1] compared to 4.2 ± 0.9 mg L[−1]) than those annual measurements given by Ward et al.[32]. Our estimate to the marine DBC pool does not include any fossil fuel aerosol inputs, given that they apparently have only local, ephemeral influence on riverine DBC as fossil fuel signals are removed downstream. Thus, we used the modern DBC values at Almeirim to estimate mass balances between river and the marine DBC pools.

The predominately modern DBC Δ[14]C signature (−46 ± 15‰) suggests that DBC is not stored in short-term intermediate reservoirs within the Amazon River, at least during low-flow conditions. Using our measurements as the riverine endmember, a mass balance calculation assuming a background marine DBC pool (−945 ± 6‰) and a freshwater river influence of 20%[13], yields a Δ[14]C value for Amazon plume marine DBC of −765 ± 35‰, which agrees within error of the measured BC value (−727 ± 44‰; 10,400 ± 1300 [14]C years) in ultra-filtered DOC[13]. This suggests that modern DBC is exported from the Amazon River, regardless of hydrological state, where it contributes to oceanic DBC.

The marine DBC pool is older and smaller than expected given the riverine DBC signature and input rates (23,000[14]C years and only 12–14 Tg[14,53]), suggesting that DBC loss by processes, such as UV oxidation and sedimentation, must be important[48]. For example, we would expect a larger concentration of DBC in the Amazon River plume (3.3 ± 0.8 µM) than observed (0.3 µM, BC in ultra-filtered DOC[13]). Thus, along with DOM, a portion of DBC may be decomposed along the river–ocean interface by photochemical degradation both at the mouth–plume interface, and where the clearwater tributaries mix[54]. Yet, not all DBC is removed at this interface because a smaller signal of recalcitrant DOM, including DBC, persists to the river–mouth interface at the Atlantic Ocean[52,54]. Medeiros et al. (2015) found that dilution was the primary factor influencing DOM variability in the plume (~60% of total variability), while biodegradation, photo-oxidation, and phytoplankton production all played a smaller, but relevant role (~5–8% of variability each)[52].

There are at least two marine DBC pools—younger pool cycling on centennial timescales and a stable pool with residence times >10[5] year timescales[14,17]. Another modern and semi-labile DBC pool from rivers may contain recent biosphere-derived BC that has not accumulated on land due to short continental

residence times, as observed here. Our study suggests that DBC is predominantly of modern origin in the Amazon River, but some river DBC does survive to accumulate in the ocean. DBC dynamics appear to be decoupled from those of DOC in this study. Labile DBC may be decomposed along the river–ocean continuum, particularly in clearwater tributaries and beyond the plume where photochemical degradation may occur, while contributions from atmospheric (aerosol) BC deposition may be locally important, but rapidly removed or diluted during downstream transport. Future work is needed to understand the underlying causes of this de-coupling, seasonal and interannual variability in isotopic values and fossil fuel aerosol DBC and PBC contributions in the Amazon during high and low flows.

## Methods

**Sampling and site locations.** DBC samples were collected along the Amazon River in November 2015, during the dry season. The 12 sites were accessible by boat. We sampled in the dry season because the flood plain extents are the lowest in November and December[55]. During the dry season, the hydrological connectivity of the Amazon decreases, as overbank flow paths, and lakes serving reservoirs (for flood waters, rainfall, and saturated water table seepage) decrease.

Surface water (at 1 m) samples were collected using deployed 2 L Niskin sampling cylinders from a small ship. The Niskin was deployed and flushed with river water before samples were collected. Samples were collected in the middle of the river when possible. Samples were also collected away from boat traffic, and all sampling was conducted in a covered area on the boat. First, all water from the five Niskin deployments were combined in acid-washed container, rinsed with river water before being filled. Samples were filtered using combusted GF/F filters (Whatman, nominal pore size 0.7 μm) to remove POC. The filtrate river water was subsampled into acid cleaned, pre-combusted amber glass bottles (100 mL), and acidified for later DOC concentration and $\Delta^{14}C$ analysis. Measurements were taken from distinct samples. To obtain enough DOC for DBC analysis, DOC (ranging from 8.8–10 L) was loaded onto solid phase extraction (SPE) cartridges. Briefly, DOC was concentrated according to Dittmar et al.[56], as DOC was acidified to pH 2 (around 20 mL high purity HCl) and samples were concentrated via SPE on PPL resin (5 g, Agilent #12256087) immediately after collection. 10 L of DOC (placed in two acid-cleaned plastic bottles) was loaded by gravity filtration onto a manifold consisting of four SPE cartridges. DOC was loaded onto a manifold of four SPE cartridges to generate four duplicates per site, stored at −25 °C and shipped back to the University of Zurich before elution. Salt was removed by rinsing all SPE cartridges with 0.01 mol L$^{-1}$ HCl. The PPL cartridges were dried under a gentle stream of N$_2$, DOC was eluted using 30 mL of high purity methanol. The SPE-DOC recovery (compared to DOC concentrations) was 63 ± 5%.

The molecular DOM composition was analyzed from SPE-DOC samples via 15T FT-ICR-MS (solariX XR, Bruker Daltonics) with electrospray ionization (ESI) in negative mode as described in ref.[57]. Briefly, methanol extracts were diluted 1:1 (v/v) in ultrapure water to 5 μg C L$^{-1}$. Mass spectra were collected over 200 scans, with an ion accumulation time of 0.15 s, in a range of 92–2000 m/z. Molecular formulae were calculated for masses with relative intensities above the method detection limit[58] allowing C$_{1-130}$H$_{1-200}$O$_{1-50}$N$_{0-4}$S$_{0-2}$P$_{0-1}$. Intensity-weighted averages of aromaticity (modified aromaticity index, AI$_{mod}$[59,60], number of atoms per molecular formula (carbon, C, hydrogen, H, oxygen, O, nitrogen, N, sulfur, S, and phosphorus, P), and molar ratios (hydrogen-to-carbon, H/C and oxygen-to-carbon, O/C) were calculated for each sample by considering the peak intensity of each assigned molecular formula as described in ref.[57]. Principal coordinate analysis (PCoA) was performed on a Bray–Curtis dissimilarity matrix of the normalized peak intensities of PCA molecular formulae (AI$_{mod} \geq 0.67$, C ≥ 15) post-hoc fitted with environmental data and molecular parameters using the envfit function of the *vegan* package[61] within the R statistical platform[62].

**Dissolved black carbon concentrations.** DBC was extracted from the SPE-DOM methanol extracts using the BPCA benzene polycarboxylic acid (BPCA) method[33,63]. Briefly, SPE-DOM extracts were dried and lyophilized for 24 h. Concentrated nitric acid was added to the sample in a quartz pressure digestion chamber at 170 °C for 8 h to produce BPCAs. After digestion, the solution was filtered, lyophilized, and re-dissolved in methanol. BPCAs were separated and collected on a preparative liquid chromatography using an Agilent 1290 infinity HPLC system equipped with a 2.7 μm Agilent Poroshell 120 C-18 column. A reverse phase analytical C-18 column (Agilent, 2.7 μm) was used with two mobile phases of pH 2 Milli-Q (1.7% H$_3$PO$_4$) and acetonitrile (>99.98% Scharlau, F$^{14}$C < 0.004). Quantification of BPCAs were made from seven-point calibration curves (2–200 ng μL$^{-1}$) using commercially available BPCA standards including penta-carboxylic acid (Aldrich S437107) and hexacarboxylic acid (Aldrich M2705) to quantify the BPCAs measured from peak areas obtained from the diode array detector (60 mm path length) chromatographs. A BC recovery factor of 23.2 ± 0.4% was used for the conversation of BPCAs to estimate BC[64,65] for comparison with published values.

**Isotopic analysis.** For DBC $\Delta^{14}C$ analysis, B5CA and B6CA marker compounds were collected in the fraction collector of the HPLC, according to Wiedemeier et al.[33]. The B2CA marker compounds were not collected, because they may also be derived from aromatic compounds of non-combusted origin (e.g. lignin). Dead (F$^{14}$C = 0.003 ± 0.001) and modern (F$^{14}$C = 1.149 ± 0.004) wood char black carbon standards during the entire BPCA procedure were used to evaluate the extraneous, or non-sample blank carbon added to samples during chemical processing[66] (Supplementary Fig. 8). BPCAs in the vials were dried under a gentle stream of ultra-high purity nitrogen on a heating plate (70 °C) for 3 h, and stored at −25 °C until wet-oxidation to CO$_2$ gas for isotopic analyses[67,68].

For DOC $\Delta^{14}C$ analysis, and DOC concentration measurements we used a wet-chemical oxidation following Lang et al.[67] and Wiedemeier et al.[33]. Briefly, 8 mL of acidified DOC (pH 2, HCl) was transferred into pre-combusted borosilicate glass Exetainer (septa sealed 4.5-mL exetainers vials from Labco Limited, UK) vials, frozen and freeze dried overnight (with pre-combusted aluminum covers). Samples were re-dissolved in Milli-Q water to a final volume of 4 mL. Milli-Q blanks, modern and dead standards (sucrose and phthalic acid) were used to evaluate the extraneous carbon added during the wet chemical oxidation procedure.

BPCAs and DOC were then converted to CO$_2$ using the wet oxidation procedure for $\Delta^{14}C$ measurement using a gas ion source AMS[33,67,68]. Briefly, 30 μg C BPCA samples, 1 mL of purified sodium persulfate and 3 mL of Milli-Q water (for a total volume of 4 mL) were transferred to gas-tight borosilicate Exetainer vials. All samples were purged with ultra-high purity helium (100 mL min$^{-1}$, 8 min to remove atmospheric CO$_2$), and oxidized to CO$_2$ in a heating block (95 °C, 1 h). Radiocarbon measurements of DBC and DOC were made on the Mini Carbon Dating System (MICADAS) Accelerator Mass Spectrometer coupled to a carbonate system modified with a needle to sparge sample solutions at the ETH Zurich Ion Beam Laboratory. DBC samples were corrected for extraneous carbon according to Hanke et al.[68] (Supplementary Fig. 8, Supplementary Table 2) and all samples were corrected using phthalic acid and sucrose standards. Radiocarbon is reported in F$^{14}$C and then converted to $\Delta^{14}C$ (‰) using the year of sampling. For DOC concentration measurements, the CO$_2$ gas measured by the AMS was normalized to the volume of DOC used (8 mL). DOC concentration measurements were better than 0.03% based on standards. Radiocarbon measurements were corrected for isotopic fractionation via $^{13}C/^{12}C$ isotopic ratios. $^{14}C$ data are reported as $\Delta^{14}C$ values (‰).

For $\delta^{13}C$ of SPE-DOC, we also used the wet-oxidation procedure to prepare samples, following Lang et al.[69] at ETH Zurich. The SPE-DOC stable organic isotopic composition ($\delta^{13}C$, ‰VPDB) was measured on the headspace CO$_2$ with a Gas Bench II on-line gas preparation and introduction system (Thermo Fisher Scientific, Bremen, Germany). Samples were prepared 1 day before processing for $\delta^{13}C$. The Gas Bench II is equipped with a CTC autosampler (CTC Analytics AG, Zwingen, Switzerland) and coupled to a ConFlo IV interface and a Delta V Plus mass spectrometer (both Thermo Fisher Scientific). For each sample, three reference gas peaks are measured and the sample gas is introduced four times into the mass spectrometer. The blank was 0.2 μg C. Corresponding $\delta^{13}C$ values were determined to an accuracy of ±0.1‰ based on phthalic acid and sucrose standards. BPCA $\delta^{13}C$ could not be measured because sample sizes were too small.

**Gauge measurements and mass balance hydrological approach.** We relied on a combination of gauge records, direct measurements during November 26–December 11, 2014 and mass balance approximations to estimate discharge in November 2015. Daily gauge data were taken from the network Agência Nacional de Águas (ANA), from the data portal Séries Históricas de Estações for Manacapurú and Óbidos (http://www.snirh.gov.br/hidroweb/publico/medicoes_historicas_abas.jsf). Rio Madeira and Alter do Chão (Rio Tapajós) and Rio Negro were derived from upstream ANA stations. Almeirim was measured directly by ADCP by the TROCAS project[70]. The remaining stations were done by mass balance approximations (Supplementary Table 2). Calculated DOC loads (Tg year$^{-1}$) used these discharge measurements in Eq. (1), where DOC is the DOC concentration (g L$^{-1}$) of 4.2 ± 0.9 mg L$^{-1}$ made at the river mouth[32] multiplied by 1000 (in units of L m$^{-3}$), 60 s, 60 min, 24 h, and 365 days, and the discharge Q (m$^3$ s$^{-1}$) of 203,000 m$^3$ s$^{-1}$ from Ward et al.[32].

$$(DOC)(1000)(60*60*24*365)*Q \qquad (1)$$

$$(DOC*DBC\%)(1000)(60*60*24*365)*Q \qquad (2)$$

We calculated DBC flux in Tg year$^{-1}$ using Eq. (2), using the same DOC concentration (g L$^{-1}$) and discharge (m$^3$ s$^{-1}$) from Ward et al.[32]. To determine the flux to the ocean, we used the range of DBC% during low flow conditions of 7.2 ± 0.5% (this study, from Amazonas-Almeirim station 1) to high flow conditions of 9.9 ± 1.0% (from 2010 at Macapa[52]). Although this is the first DBC Amazon flux estimate, this measurement includes DBC measurements during the dry season (2015) and wet seasons (2010)[52]. For example, the DOC concentration at Amazonas-Almeirim (2.5 ± 0.3 mg L$^{-1}$) measured during this time period (2015) is much lower than the average concentration (4.2 ± 0.9 mg L$^{-1}$) made at the river mouth[32]. Future constrains on this first estimate can be made

by sampling multiple cross channel locations, and at multiple depths, during all seasons across several years.

Catchment boundaries corresponding to each sample site were determined using a high-resolution (~500 m) stream drainage direction map for the Amazon River basin[71]. Supplementary Fig. 1c, d was created with ArcMap 10.6, relief shade from Natural Earth dataset (http://www.naturalearthdata.com), catchments derived from CAMREX (Carbon in the Amazon River Experiment) dataset (https://daac.ornl.gov/LBA/guides/CD06_CAMREX.html) and runoff from GSCD[72] (https://water.jrc.ec.europa.eu/).

**Land use, urbanization, and fire history.** Land cover (from Modis 2012, at 15 arc minutes resolution https://modis.gsfc.nasa.gov/data/dataprod/mod12.php), urban areas (Global Rural-Urban Mapping Project GRUMP)[73] and burned area (Global Fire Emissions Database, GFED4s) for each sample was integrated over the cell area (in km$^2$) for the sample. We used an estimate of fire history from satellite observations from GFED4s (available http://www.globalfiredata.org/index.html for the past decades (1997–2015))[74]. The parameters were intergraded over the corresponding sample location's catchment area (Supplementary Fig. 4, Supplementary Table 3) which was determined by using a high-resolution (~500 m) stream drainage direction map for the Amazon River basin[71] (Supplementary Fig. 1c). All panels in Supplementary Fig. 4 were created with ArcMap 10.6, with relief shade from Natural Earth dataset (http://www.naturalearthdata.com) and catchments derived from CAMREX (Carbon in the Amazon River Experiment) dataset (https://daac.ornl.gov/LBA/guides/CD06_CAMREX.html).

**Atmospheric deposition of black carbon aerosols.** Aerosol BC deposition was modeled for 2015 in South America using the UK Met Office Hadley Centre Global Environment Model version 2 earth system model (HadGEM2-ES[10,75,76]. Briefly, HadGEM2-ES represents the life cycle of aerosol species[77], including from fossil fuel and biofuel emissions and from biomass burning. BC is modeled as an internally mixed component of organic carbon[78]. Processes such as transport, mixing and deposition are represented explicitly through physically based parameterizations that have been developed and constrained using observations. HadGEM2-ES was run with standard climate resolution (1.875° × 1.25°) in the period 2009–2016. Simulations were driven by a custom biomass burning aerosol emission dataset produced for South America as discussed by Jones et al. [10]. Briefly, published carbon stock consumption factors (kg dry matter km$^{-2}$)[79] and aerosol BC emission factors (g BC kg dry matter$^{-1}$) [80] were assigned to burned area cells according to land cover type in the MODIS MCD12Q1 dataset (resolution 500 m) enhanced with information on ecoregion[81] and agricultural land use[82]. Biomass burning aerosol emissions outside of South America were taken from the Global Fire Emission Dataset (GFED) version 3.1[83]. Emissions of aerosol from fossil fuel sources followed the dataset employed in the Coupled Model Intercomparison Project phase 5 (CMIP5), based on the historical emissions[84] with regional updates following the Representative Concentration Pathway (RCP) 8.5[85,86]. The deposition rates from HadGEM2-ES were interpolated to a 0.05° grid using an empirical Bayesian kriging operation performed in the ArcGIS Geostatistical Analyst toolbox. The approach and performance of the kriging process are reported by Jones et al.[10] and summarised in the supplementary material. Rates of biomass and fossil fuel-derived aerosol deposition (kg C km$^{-2}$ year$^{-1}$) were averaged within the upstream catchments of all sampling locations on the river network (Supplementary Fig. 4, Supplementary Table 3).

Deposition rates from HadGEM2-ES were transferred to higher-resolution grids using an empirical Bayesian kriging operation performed with the ArcGIS Geostatistical Analyst toolbox. The process was completed using a log-empirical transformation, Whittle de-trended semivariogram, and search radius of 4°. The normalized mean error of the predictions from the kriging operation, when compared to the data from the native HadGEM2-ES grid, was 0.7% for total fossil fuel BC aerosol and 1.7% for biomass burning BC aerosol deposition during 2009–2016. Predictions were mapped to a 0.05° grid. The normalized root mean square error of the predictions from the kriging operation, when compared to the input data from the native HadGEM2-ES, was 5% for total fossil fuel BC aerosol and 15% for biomass burning BC aerosol deposition during 2009–2016.

Although the prevailing wind direction in Amazonia is from East to West, changes in wind direction provide an opportunity for aerosols emitted in southern Brazil to be transported northwards and deposited in the Amazon Basin[41]. This influences the distribution of BC aerosol deposition across central Brazil and leads to raised deposition rates in the south of the Amazon relative to northern regions that are further from the main population centers of the country. There may be an imprint of sources in southeast Brazil on the background deposition rates of northern export of BC from cities.

## Data availabilty
The data set will be on the PANGAEA data (www.pangaea.de) repository following publication under the username alyshacoppola (connected to the ORCIDID https://orcid.org/0000-0002-9928-2786). In addition, all data are available from the corresponding author on request. Please contact Alysha Inez Coppola (Alysha.coppola@geo.uzh.ch, or at http://alyshainezcoppola.strikingly.com/) for correspondence and material requests.

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

## Acknowledgements
We thank Michael Hilf for assistance in the laboratory. We also thank Derek Vance, Chantal Freymond, and the crew onboard the Joao Felipe (November 2015) for assistance, organization, and coordination of collecting river water samples. We thank Ilja van Meerfeld for comments on an earlier version of the manuscript and helpful discussions. We also thank Katrin Klaproth (University of Oldenburg) for support with the FT-ICR-MS analysis. We thank Lukas Wacker and the staff of ETH Laboratory of Ion Beam Physics for AMS support. We thank Madalina Jaggi for support with δ13C measurements. We thank Ben Johnson (UK Met Office) for generating the aerosol BC deposition grids by running the HadGEM2-ES model. We thank Ellen Druffel for comments on the manuscript. A.I.C. thanks Natalie Renier (WHOI Media Team) for support in making Fig. 1. A.I.C. acknowledges financial support from the University of Zurich Forschungskredit Fellowship and the Science Faculty of the University of Zurich (No. STWF-18-026). S.A. and A.I.C. acknowledges support from SNSF #200021_178768. M.W.I.S. acknowledges support for the University Research Priority Program: Global Change and Biodiversity. J.E.R. and N.D.W. acknowledges financial support from NSF DEB Grant #1256724 and FAPESP Grant #08/58089-9.

## Author contributions
A.I.C., M.W.I.S., T.D. and T.I.E. led efforts in devising the study objectives and design of the work. A.I.C. conducted concentration and isotopic measurements of DOC and DBC, analyzed data, interpreted the data and wrote the paper. G.S. collected samples. B.N.R. coordinated traveling logistics to Brazil. N.H. provided support and quality control with DBC and DOC 14C measurements and radiocarbon corrections. S.A. provided input with discussions. M.S. and T.D. added FT-ICR-MS measurements, associated data analyses and scientific interpretation. D.V., J.E.R. and N.D.W. provided support for hydrology context. N.D.W. and J.E.R. provided river gauge data, hydrological expertize, input on the interpretation of the data and substantively revised the manuscript. D.V. provided discharge, land use, and urbanization catchment-specific data. M.W.J. provided outputs of atmospheric BC deposition from the HadGEM2-ES model and input for the interpretation of the data. All authors provided comments in constructing the final version of the manuscript.

## Additional information

**Competing interests:** The authors declare no competing interests.

