## [Peer Review File · Nature Communications]

Reviewers' comments:

Reviewer #1 (Remarks to the Author):

This paper presents a series of radiocarbon and molecular marker analyses for dissolved organic and black carbon from along the Amazon mainstem and major tributaries in the middle reaches of the river. The analyses are sophisticated and well explained and the dataset is very valuable. However, I cannot recommend publication of the manuscript because of a couple of potentially major errors (which unfortunately may be due to simple typos).

Specifically – the argument of the paper is that a significant fraction of the dissolved black carbon (DBC) is derived from fossil fuel combustion near major towns. This focuses on four low radiocarbon samples described in the text at line 147 as at four sites (Trombetas-Oriximina, -771‰ , Amazonas-Santar.m $-431\pm 15\text{‰}$, Obidos $-658\pm 6\text{‰}$, and downstream of Manaus $-720\pm 9\text{‰}$). From supplementary Figure 1 and Supplementary table 1 these are sites 3,4,5 and 10. The problem then is that site 5 has a DBC C14 of $+11\text{‰}$ not -771‰ . It is site 2 on the Tapajos river that is -771‰ . This site is remote from cities (I think) and also drains the largest savanna area of any of the catchments, hence there is likely a long-term large and local black carbon source. The problem therefore may be a simple typo or may flow through all the analysis and interpretation in the paper, and I can't tell which is the case. Hence I am unable to complete the review.

In any resubmission I think the paper would benefit from the following:

- There is no mention of direct aerosol deposition into the ocean as a significant input to the ocean DBC pool in the introduction, which focuses solely on riverine inputs. Yet a large part of the paper is devoted to aerosol inputs into rivers. I think it would be worth elaborating on direct aerosol input into the ocean in the introduction.
- I think the use of 2015 burnt area as the parameter to test against DBC concentrations is probably not appropriate and would be better to use burnt area averaged over a much longer timeframe instead (or in addition)
- A location map including the sites should really be in the main text, and site numbers should be used not site names to avoid confusion
- Figure 2a says that Aerosol BC deposition was modelled, and refers to supplementary information for details. I cannot see any supplementary information that details how this modelling was done? Only the fossil fuel aerosol data is shown in figure 2a in the panel b but the legend refers to all aerosol BC (which is it?). The biomass BC aerosol numbers are provided in supplementary table 3 and I would have thought should have also been plotted in a figure even if only for supplementary display as the totals are similar in magnitude to the fossil fuel derived BC

Reviewer #2 (Remarks to the Author):

Thank you for giving me this opportunity to review the manuscript (NCOMMS-19-08714) by Alysha Coppola et al. I know Alysha has been working on black carbon since her Ph.D. at UCI. And she has several good papers published on the subject. After reading this manuscript, here is my thought.

Dissolved black carbon (DBC) has been a very interesting topic recently and people in several labs including myself are working on it. This is a well written paper and it brings a very good question regarding the radiocarbon ages of DBC we report. First of all, there has been no a single internationally accepted standard method used for separating BC from complex natural samples. Chemothermal oxidation (CTO) and BPCA (benzene polycarboxylic acid) are the two commonly used methods and both have advantage and disadvantages. That's OK no matter what people say. It would be interesting to see some comparison between the two methods for the same samples.

The most interesting point or concern I have is why the radiocarbon ages of DBC varied so largely in the same Amazon River system. The ages of DBC are not related to their concentrations at all. The authors made some explanations but I am not fully convinced. Aerosol deposition could certainly bring some BC from fossil fuel combustion but I don't think this process could make such big changes for DBC ages in a same river. Also, most aerosol BC tends to stay in particulate form, not dissolved. What are the reasons? We don't know. As the authors said "Line 176: The sharp isotopic variability within DBC, brings up more questions than answers".

In the last two year, my students sampled four large rivers in China and three small mountainous rivers in Taiwan and we used the same solid-phase-extraction (SPE) method to concentrate DBC and measured radiocarbon compositions of DBC. However, we found that the ^{14}C ages of DBC in all these rivers are young, younger than the ages of bulk DOC, consistent with our initial report (Wang et al., 2016, GBC). Also, we concentrated DBC from the East China Sea and the North Pacific waters, and we still see DBC ages are relatively younger than DOC. This is certainly a big disagreement with the very old DBC ages (18000-20000 yrs) in the ocean as reported by Coppola and Druffel (2016, GRL) and Ziolkowski and Druffel (2010, GRL). We will write a manuscript this year and hope to add more interesting data and discussion to this hot topic.

I support the publication of this manuscript in Nature Communication after minor revision, especially for Figure 1d that is presented in a very confusing way. The results are interesting even without convince explanation. I think this is the science discovery all about. The discussion and controversy will continue on this interesting subject.

Reviewer #3 (Remarks to the Author):

During the last two decades, the attention of blank carbon (BC) have been arisen quickly in the field of organic geochemistry. The research directions have been driven from the sediment/solid BC to the dissolved BC, from hydrological and dynamic studies to molecular characteristic and radioactive signature studies. The manuscript here presented a very good example of a comprehensive and well-designed research project, using multiple state-of-the-art analytical techniques to tell a story of DBC in a unique environment. There is no doubt about how important the Amazon is to our ecosystems. The understanding of the cycling of DBC and DOC in the Amazon River is of a global significance considering it is the largest terrestrial DOM source to the ocean. This manuscript presented some interesting results. For instance, the DBC and DOC are decoupled in this unique case, unlike many other reports that DBC and DOC are correlated. It also presented the variation of DBC $\Delta^{14}\text{C}$ signature along the riverine transact, and the relative young radiocarbon age for the exported DBC from Amazon river. The paper gave out some good explanations for some of the results they observed, such as hydrological reason for the low estimates of DBC and DOC, different BC sources may contribute the variation of DBC carbon signature and explain the decoupling between DBC and DOC, and implies of significant removal processes of DBC in the coastal plume. What's interesting is that this manuscript and this research project actually lead to more research questions. As stated by the authors, more follow up studies will be needed, including a larger temporal and spatial scale sampling, in order to enlarge the dataset with more statistical meaning. In general, the study was well designed, the manuscript was well written and all the data were solid and well presented. The environmental importance of the data is of global significance, fitting the scope of "Nature Communication". Thus, I recommend accepting this manuscript with minor corrections.

Below are my comments for minor correction:

1. In general, I think the figures in the manuscript could be clearer and easier to follow if there is a consistent naming system for the sites among figures. For example, the sites in Fig. 1 are labeled by name/location, sites in Fig. 2 are not labeled, sites in Fig. 3 are labeled with numbers. It's very confusing to cross reference the sites between the figures, especially for Fig. 2. I suggest putting

numbers on Fig. 1 in front of the location, and Fig.2 and 3 can use the same numbers.

2. I understand that the authors chose to sample in the dry season in order to simplify the hydrological impacts and sources. However, I think a follow up study in the wet season will be appropriate since numerous more DOM and DBC will be exported from Amazon River in the wet season. A comparison between wet and dry season will be very interesting to see any molecular characteristics and radiocarbon signature difference. It can be a good follow up paper to be published in the same journal.

3. Line 146-147: From Fig. 1d, the lowest $\Delta 14C$ value for DBC are found in Manaus, Óbidos, Santarém and Alter do Chão. Line 146 has site Oriximina as one of the four. Was this a typo here, or a mislabel in Fig. 1d?

4. Line 180-184: When I first read this paragraph, it is hard for me to believe that 46-68% of DBC for some sites are originated from aerosol. My questions is can the low radiocarbon age is caused by the sediment? Is there any carbon dating for the sediment? However, I noticed that the DOC carbon age is not old for those sites, so DBC/DOC exported from the sediment should not be the reason. The authors can add a few sentences here to clarify that sediment age should not be the reason for the distinctively low carbon 14 dating.

5. Line 214-219: Have you try to analyze aerosol samples directly and it's leachate in FT-ICR-MS in order to see if you have any unique signature formulae?

6. Line 234-235: The statement here contradict with the previous explanation that for some sites, fossil fuel aerosol is a significant input of DBC.

7. Line 336: black carbon, typo

8. Line 391: $((DOC \text{ g L}^{-1})(1000 \text{ L m}^{-3})(60 \text{ s} * 60 \text{ min} * 24 \text{ hr} * 365 \text{ days})) * (Q \text{ m}^3 \text{ s}^{-1} 391)$

Sincerely,

Yan Ding, PhD

Technical Director
CACE-Nutrient Analysis Core Facility
Southeast Environmental Research Center (SERC)
Institution of Water and Environment (InWE)
Florida International University, Modesto Maidique Campus

Reviewers' comments:

Reviewer #1 (Remarks to the Author):

This paper presents a series of radiocarbon and molecular marker analyses for dissolved organic and black carbon from along the Amazon mainstem and major tributaries in the middle reaches of the river. The analyses are sophisticated and well explained and the dataset is very valuable. However, I cannot recommend publication of the manuscript because of a couple of potentially major errors (which unfortunately may be due to simple typos).

Specifically – the argument of the paper is that a significant fraction of the dissolved black carbon (DBC) is derived from fossil fuel combustion near major towns. This focuses on four low radiocarbon samples described in the text at line 147 as at four sites (Trombetas-Oriximina, -771‰, Amazonas-Santar.m -431±15‰, Obidos -658±6‰, and downstream of Manaus -720±9‰). From supplementary Figure 1 and Supplementary table 1 these are sites 3,4,5 and 10. The problem then is that site 5 has a DBC C14 of +11‰ not -771‰. It is site 2 on the Tapajos river that is -771‰. This site is remote from cities (I think) and also drains the largest savanna area of any of the catchments, hence there is likely a long-term large and local black carbon source. The problem therefore may be a simple typo or may flow through all the analysis and interpretation in the paper, and I can't tell which is the case. Hence I am unable to complete the review.

We apologize for the discrepancy and typo. As the other reviewers also pointed out- the labeling system was confusing so we replaced the text for consistency, referring to sampling sites with both site location and numbers in Lines 146-148. Values in Figure 1 D and Table S1 were correct (now Fig S3). We also replaced Figure 1D to more accurately show the catchment area of the sampling locations to highlight that site 2 on the Tapajós River drains the largest savanna area (Fig S1).

In any resubmission I think the paper would benefit from the following:

- There is no mention of direct aerosol deposition into the ocean as a significant input to the ocean DBC pool in the introduction, which focuses solely on riverine inputs. Yet a large part of the paper is devoted to aerosol inputs into rivers. I think it would be worth elaborating on direct aerosol input into the ocean in the introduction.

Thank you for this comment. We agree and have incorporated an overview of aerosol BC in Lines 61-63, with “Atmospheric deposition is another major pathway in which BC reaches rivers (after mobilization from the landscape)⁹ and the ocean (1.8-10 Tg yr⁻¹)^{10,11}.”

In the introduction (Lines 101-103), we also incorporated more information about aerosols specific to the Amazon River basin with “The Amazon Basin transitions between pristine forest and urban-influenced aerosol polluted plumes due to rapid developments in energy, agriculture expansion and deforestation²⁸.”

I think the use of 2015 burnt area as the parameter to test against DBC concentrations is probably not appropriate and would be better to use burnt area averaged over a much longer timeframe instead (or in addition)

Yes we agree. This was also a mistake. Burned area was estimated from GFED4s from 1997-2015, but the figure was not updated. We included an updated figure. We apologize, this was also a typo as explained in the supplementary materials lines 388-390 with “We used an estimate of fire history from satellite observations from GFED4s (available <http://www.globalfiredata.org/index.html> for the past decades (1997–2015).” Supplementary Figure 2b was updated to show the entire range in the catchment.

- A location map including the sites should really be in the main text, and site numbers should be used not site names to avoid confusion

These changes have been implemented. We agree and have gotten professional help to create Figure 1 to include the location map. We agree with your suggestion and were consistent to use site numbers instead of station names throughout the text to avoid confusion.

Figure 1. Sample locations for this study plotted as distance to the river mouth (in km) with the river and nearest city. The shades of yellow to red represents modern to ancient DBC $\Delta^{14}\text{C}$ values. Grey indicates no $\Delta^{14}\text{C}$ data at site 8. DBC concentrations are shown by the height of the columns in $\mu\text{g BC L}^{-1}$.

Figure 2a says that Aerosol BC deposition was modelled, and refers to supplementary information for

details. I cannot see any supplementary information that details how this modelling was done? Only the fossil fuel aerosol data is shown in figure 2a in the panel b but the legend refers to all aerosol BC (which is it?). The biomass BC aerosol numbers are provided in supplementary table 3 and I would have thought should have also been plotted in a figure even if only for supplementary display as the totals are similar in magnitude to the fossil fuel derived BC

We have added supplementary information with how this modeling was done with its own subheading for clarity from Lines 393-436. The legend for Figure S9 was replaced with “Fossil fuel BC 2015 annual average deposition rates for the upstream catchment of each sampling location.” We agree that the previous text in the legend is misleading because I was referring to the fact that aerosol BC was integrated over the entire sub-catchment for these sampling locations. The totals are similar in magnitude to the fossil fuel BC deposition rates (Table S3, Fig S4). In general, we limited the discussion of aerosols in the main text and moved aerosols as a possibility to explain the 4 points to the discussion.

Reviewer #2 (Remarks to the Author):

Thank you for giving me this opportunity to review the manuscript (NCOMMS-19-08714) by Alysha Coppola et al. I know Alysha has been working on black carbon since her Ph.D. at UCI. And she has several good papers published on the subject. After reading this manuscript, here is my thought.

Dissolved black carbon (DBC) has been a very interesting topic recently and people in several labs including myself are working on it. This is a well written paper and it brings a very good question regarding the radiocarbon ages of DBC we report. First of all, there has been no a single internationally accepted standard method used for separating BC from complex natural samples. Chemothermal oxidation (CTO) and BPCA (benzene polycarboxylic acid) are the two commonly used methods and both have advantage and disadvantages. That's OK no matter what people say. It would be interesting to see some comparison between the two methods for the same samples.

The most interesting point or concern I have is why the radiocarbon ages of DBC varied so largely in the same Amazon River system. The ages of DBC are not related to their concentrations at all. The authors made some explanations but I am not fully convinced. Aerosol deposition could certainly bring some BC from fossil fuel combustion but I don't think this process could make such big changes for DBC ages in a same river. Also, most aerosol BC tends to stay in particulate form, not dissolved. What are the reasons? We don't know. As the authors said “Line 176: The sharp isotopic variability within DBC, brings up more questions than answers”.

In the last two year, my students sampled four large rivers in China and three small mountainous rivers in Taiwan and we used the same solid-phase-extraction (SPE) method to concentrate DBC and measured radiocarbon compositions of DBC. However, we found that the ^{14}C ages of DBC in all these rivers are young, younger than the ages of bulk DOC, consistent with our initial report (Wang et al., 2016, GBC). Also, we concentrated DBC from the East China Sea and the North Pacific waters, and we still see DBC ages are relatively younger than DOC. This is certainly a big disagreement with the very old DBC ages (18000-20000 yrs) in the ocean as reported by Coppola and Druffel (2016, GRL) and Ziolkowski and Druffel (2010, GRL). We will write a manuscript this year and hope to add more interesting data and discussion to this hot topic.

We highlighted in Lines 377-379 that we really need more studies in the future addressing this conundrum between river and ocean DBC with “Future studies need to address the conundrum between the old DBC ages in the marine and riverine DBC pools.” The problem is there is patchiness of point sources and only one DBC $\Delta^{14}\text{C}$ river study from Wang et al., (2016) to address these questions. There are uneven distributions of $\Delta^{14}\text{C}$ values between dissolved, colloidal and particulate pools with fast exchange between phases through desorption, absorption, flocculation and degradation. Also there are different analytical windows of qualitative DBC data compared to the compound specific $\Delta^{14}\text{C}$ of BPCAs. This and Wang et al., (2016) data provides the data to build upon

to identify key mechanisms responsible at these aquatic continuums and understand how DBC cycles.

I support the publication of this manuscript in Nature Communication after minor revision, especially for Figure 1d that is presented in a very confusing way. The results are interesting even without convince explanation. I think this is the science discovery all about. The discussion and controversy will continue on this interesting subject.

Thank you for your valuable feedback on the manuscript. We have changed Figure 1 to make the data less confusing by including site locations, upstream catchment areas and DBC 14 values.

Reviewer #3 (Remarks to the Author):

During the last two decades, the attention of blank carbon (BC) have been arisen quickly in the field of organic geochemistry. The research directions have been driven from the sediment/solid BC to the dissolved BC, from hydrological and dynamic studies to molecular characteristic and radioactive signature studies. The manuscript here presented a very good example of a comprehensive and well-designed research project, using multiple state-of-the-art analytical techniques to tell a story of DBC in a unique environment. There is no doubt about how important the Amazon is to our ecosystems. The understanding of the cycling of DBC and DOC in the Amazon River is of a global significance considering it is the largest terrestrial DOM source to the ocean. This manuscript presented some interesting results. For instance, the DBC and DOC are decoupled in this unique case, unlike many other reports that DBC and DOC are correlated. It also presented the variation of DBC $\Delta 14C$ signature along the riverine transact, and the relative young radiocarbon age for the exported DBC from Amazon river. The paper gave out some good explanations for some of the results they observed, such as hydrological reason for the low estimates of DBC and DOC, different BC sources may contribute the variation of DBC carbon signature and explain the decoupling between DBC and DOC, and implies of significant removal processes of DBC in the coastal plume. What's interesting is that this manuscript and this research project actually lead to more research questions. As stated by the authors, more follow up studies will be needed, including a larger temporal and spatial scale sampling, in order to enlarge the dataset with more statistical meaning. In general, the study was well designed, the manuscript was well written and all the data were solid and well presented. The environmental importance of the data is of global significance, fitting the scope of "Nature Communication". Thus,

I recommend accepting this manuscript with minor corrections.

Below are my comments for minor correction:

1. In general, I think the figures in the manuscript could be clearer and easier to follow if there is a consistent naming system for the sites among figures. For example, the sites in Fig. 1 are labeled by name/location, sites in Fig. 2 are not labeled, sites in Fig. 3 are labeled with numbers. It's very confusing to cross reference the sites between the figures, especially for Fig. 2. I suggest putting numbers on Fig. 1 in front of the location, and Fig.2 and 3 can use the same numbers.

Thank you for your suggestions on how to improve Figure 1. We agree, and these changes have substantially improved the accessibility of the data to relate it to the manuscript text. We replaced it with your suggestions. We put site numbers on Figure 1 and used the same site numbers in Figure 2.

2. I understand that the authors chose to sample in the dry season in order to simplify the hydrological impacts and sources. However, I think a follow up study in the wet season will be appropriate since numerous more DOM and DBC will be exported from Amazon River in the wet season. A comparison between wet and dry season will be very interesting to see any molecular characteristics and radiocarbon signature difference. It can be a good follow up paper to be published in the same journal.

Yes, we thoroughly agree with reviewer #3's comments. To include the seasonality, we increased our estimation of fluxes by including values from Mederios et al., (2015) which measured DBC at the river plume during high flow. In this way, we report a range from low flow (this study) to high flow (Mederios et al., 2015). We addressed this comment in Lines 327-327 with, "Based on the average annual Amazon River DOC export (22 to 27 Tg) and a range of DBC/DOC % values at low ($7.2 \pm 0.5\%$, at Almeirim site 1, S Table 2) and high flow (DBC/DOC% of at Macapa $9.9 \pm 1.0\%$ ⁵⁶), we estimate an annual flux of DBC of 1.5 to 2.7 Tg from the Amazon River basin to the Atlantic Ocean." However, we think that the current study merits publication as is and that the comparison between seasons is focus of future studies along the river and plume.

3. Line 146-147: From Fig. 1d, the lowest $\Delta^{14}\text{C}$ value for DBC are found in Manaus, Óbidos, Santarém and Alter do Chão. Line 146 has site Oriximina as one of the four. Was this a typo here, or a mislabel in Fig. 1d?

Yes this was a typo and was corrected. We apologize for the confusion. Sites 2, 3, 4 and 10 had low ^{14}C values. We replaced site names with numbers throughout the text for consistency. Figure 1D and the table was correct, it was a typo in the text due to labeling by sampling locations. We believe it reads much easier now, due to building this consistency with naming.

4. Line 180-184: When I first read this paragraph, it is hard for me to believe that 46-68% of DBC for some sites are originated from aerosol. My questions is can the low radiocarbon age is caused by the sediment? Is there any carbon dating for the sediment? However, I noticed that the DOC carbon age is not old for those sites, so DBC/DOC exported from the sediment should not be the reason. The authors can add a few sentences here to clarify that sediment age should not be the reason for the distinctively low carbon 14 dating.

It is perhaps possible that old black carbon in the particulate phase could enter the dissolved phase via desorption. It's possible sorption and desorption between particulate and dissolved phases are indeed possibly playing a role in isotopic variability as we now note in the text.

We decided to move the discussion of the low values to the SI material, and we offer 3 possible explanations instead in the main text to explain the low values, in Lines 158-170 with "There are several potential explanations for the presence of these 4 strongly ^{14}C -depleted riverine DBC values. For example, there is an urban anthropogenic aerosol plume 8-70 km downwind of Manaus from regional urbanization⁴¹ that could contribute to the low $\Delta^{14}\text{C}$ DBC values observed at site 10 (20km downwind)⁴². A second possible explanation for low DBC $\Delta^{14}\text{C}$ values in mainstem sites just downstream of tributaries (e.g., Santarém) is that the collected water represented an impartial mixture of tributary and mainstem water considering that tributaries in the Amazon have been shown to be poorly mixed up to 100 km downstream of their confluence.⁴³ Additionally, although past measurements show that suspended POC in the lowland rivers studied here have modern radiocarbon signatures similar to DOC⁴⁰, PBC in the Amazon River has a $\Delta^{14}\text{C}$ value of $-386 \pm 43\%$ ($3,900 \pm 770$ ^{14}C yrs)⁷. This suggests that the old DBC values observed here may have some contribution derived from sedimentary material desorbed into the dissolved phase (see SI for further discussion)."

We sampled during one of the driest seasons in the Amazon basin following a strong El Niño³⁶ to capture DBC at low flow and to limit complexities associated with dissolved-particulate phase interactions and floodplain exchange.

However, similar to bulk DOC, both coarse and fine bulk particulate organic carbon has a modern age near these lowland sampling sites according to Mayorga et al., (2005). We added the following statement (Lines 166-168), " Additionally, although past measurements show that suspended POC in the lowland rivers studied here have modern radiocarbon signatures similar to DOC⁴⁰, PBC in the Amazon River has a $\Delta^{14}\text{C}$ value of $-386 \pm 43\%$ ($3,900 \pm 770$ ^{14}C yrs)⁷. This suggests that the old DBC values observed here may have some contribution derived from sedimentary material desorbed into the dissolved phase."

5. Line 214-219: Have you try to analyze aerosol samples directly and it's leachate in FT-ICR-MS in order to see if you have any unique signature formulae?

Unfortunately we did not sample aerosols in 2015 directly, nor it's leachate. We included this statement in Lines 172-176 "However, future field, laboratory and modeling studies to map urban and biomass burning emissions at different locations in the Amazon ⁵¹ paired with paired with $\Delta^{14}\text{C}$ aerosol BC, river PBC, sediments and molecular composition directly need to be conducted to fully determine the primary driver of the observed low DBC $\Delta^{14}\text{C}$ values at these four sites and address test this hypothesis and open questions."

We think that the current study merits publication as is and that the comparison between aerosols is focus of future studies.

6. Line 234-235: The statement here contradicts with the previous explanation that for some sites, fossil fuel aerosol is a significant input of DBC.

We clarified the statement in Line 214-218 with "Our estimate to the marine DBC pool does not include any fossil fuel aerosol inputs, given that they apparently have only local, ephemeral influence on riverine DBC as fossil fuel signals are removed downstream. Thus, we used the modern DBC values at Almerim to estimate mass balances between river and the marine DBC pools."

This assumption was made when upscaling to the marine DBC pool because the fossil fuel atmospheric signals in the river DBC were removed from the water column in downstream sites. We expect that sorption to suspended sediments (and subsequent flocculation) or mixing with deep waters removed these low ^{14}C values downstream. As in Lines 858-866, "Considering that each mainstem station bares the quantitative legacy of everything upstream of that point, there must be a rapid process such as sorption or decomposition driving the downstream variability in DBC $\Delta^{14}\text{C}$ values. Thus, we hypothesize that the rapid shifts from surface low DBC $\Delta^{14}\text{C}$ values (from potentially fossil fuel aerosol impacted sites) to modern $\Delta^{14}\text{C}$ values at downstream sites (sampled at 1m water depth) may be rapidly removed by transferring to larger size particulate BC pools in both suspended and benthic sediments, thus removing fossil fuel signals from the dissolved phase³⁸. Indeed, sorption and desorption is a rapid process that alters DOM composition along the Amazon River continuum⁸⁶."

7. Line 336: black carbon, typo

We corrected this to "Dead ($F^{14}\text{C}=0.003\pm 0.001$) and modern ($F^{14}\text{C}=1.149\pm 0.004$) wood char black carbon standards during the entire BPCA procedure were used to evaluate the extraneous, or non-sample blank carbon added to samples during chemical processing [*Hammes et al.*, 2007]."

8. Line 391: $((\text{DOC g L}^{-1})(1000 \text{ L m}^{-3})(60 \text{ s} * 60 \text{ min} * 24 \text{ hr} * 365 \text{ days})) * (\text{Q m}^3 \text{ s}^{-1} 391)$

Thank you we updated this change in the manuscript.

Hammes, K., et al. (2007), Comparison of quantification methods to measure fire-derived (black/elemental) carbon in soils and sediments using reference materials from soil, water, sediment and the atmosphere, *Global Biogeochemical Cycles*, 21(3), doi: 10.1029/2006gb002914.

Mayorga, E. *et al.* Young organic matter as a source of carbon dioxide outgassing from Amazonian rivers. *Nature* **436**, 538-541, doi:10.1038/nature03880 (2005).

Stallard, R. F., and J. M. Edmond (1983), Geochemistry of the Amazon: 2. The influence of geology and weathering environment on the dissolved load, *Journal of Geophysical Research: Oceans*, 88(C14), 9671-9688, doi: 10.1029/JC088iC14p09671.

Reviewer #1 (Remarks to the Author):

This version of the manuscript is much improved over the original and I believe has adequately addressed the concerns expressed in my initial review, and also those of the other reviewers. As the topic is of broad and substantial interest I am happy to recommend publication. I have some small suggestions:

- (i) the second sentence in the abstract is very long - split in two.
- (ii) just double check figure S4 - are panels D and E only 2015 or 1997-2005. Doesn't matter either way, provided the legend is correct.
- (iii) I think it could use another read for small grammar and punctuation issues - for example d13C is not formatted at line 738, Jones et al 2019 manuscript can now be fully referenced, and I suspect there are more small things I didn't pick up as I read the track changes version to make sure I could see what had changed, but the track changes version is hard to read for grammar issues.

Reviewer #2 (Remarks to the Author):

This revised manuscript has made some improvements. The figures look much better and clearer than before. Again, the main point of the study is the founding of marked differences of radiocarbon ages of dissolved black carbon (DBC) in the same Amazon River. I agree with the author that this study certainly brings up more questions than answers, and no conclusion has been reached. More studies are definitely needed to find answers for the questions. I support the publication of this manuscript in Nature Communication.

Line 143, ($\pm 55 \pm 30\%$) should be ($55 \pm 30\%$).

Reviewer #3 (Remarks to the Author):

I appreciate the authors for detailed explanations to my comments and made changes accordingly. The revised manuscript has fully addressed my previous concerns and questions, I recommend to publish this manuscript with minor corrections listed below:

1. line 126-127: station 2 is not a mainstream site according to the rest of the manuscript, please change.
2. line 128: change "to Santarem" to "at Santarem"
3. line 772-772: Figure 2 (b) does not have color-coded weight-average carbon number, please delete it from the caption.
4. line 897-900: Figure caption does not match the figure, please correct the caption.

We are delighted to read that we have satisfied the majority of the reviewer's prior comments. Thank you for your time and comments because it has substantially improved the quality of the manuscript. Regarding their comments below, we have made the changes they recommended.

REVIEWERS' COMMENTS:

Reviewer #1 (Remarks to the Author):

This version of the manuscript is much improved over the original and I believe has adequately addressed the concerns expressed in my initial review, and also those of the other reviewers. As the topic is of broad and substantial interest I am happy to recommend publication. I have some small suggestions:

(i) the second sentence in the abstract is very long - split in two.

We agree with this comment and have instead shortened the sentence to keep within the word limits, "As a large and slow-cycling component of the global carbon cycle, it is important to constrain the sources and fate of this elusive carbon pool during land-ocean transfer to determine the significance of BC as a sink of atmospheric CO₂, and to reconcile BC budgets."

(ii) just double check figure S4 - are panels D and E only 2015 or 1997-2005. Doesn't matter either way, provided the legend is correct.

We have checked and clarified that the aerosol BC deposition is only for the year corresponding for the samples were collected in 2015. We have updated this note in the Methods section for Atmospheric Deposition of Black Carbon Aerosols. We included only the 2015 emissions because aerosol deposition contributions to the catchment have a much shorter residence time (1 year), opposed to fire history, which can build up BC stocks in soils over centennial times scales.

(iii) I think it could use another read for small grammar and punctuation issues - for example d13C is not formatted at line 738, Jones et al 2019 manuscript can now be fully referenced, and I suspect there are more small things I didn't pick up as I read the track changes version to make sure I could see what had changed, but the track changes version is hard to read for grammar issues.

Thank you. We have formatted d13C and updated the Jones et al., citation. We have also made minor grammatical changes to the text (as seen in the Track Changes version).

Reviewer #2 (Remarks to the Author):

This revised manuscript has made some improvements. The figures look much better and clear than before. Again, the main point of the study is the founding of marked differences of radiocarbon ages of dissolved black carbon (DBC) in the same Amazon River. I agree with the author that this study certainly brings up more questions than answers, and no conclusion has been reached. More studies are definitely needed to find answers for the questions. I support the publication of this manuscript in Nature Communication.

Line 143, ($\pm 55 \pm 30\%$) should be ($55 \pm 30\%$).

Thank you. This was corrected to $+55 \pm 30\%$.

Reviewer #3 (Remarks to the Author):

I appreciate the authors for detailed explanations to my comments and made changes accordingly. The revised manuscript has fully addressed my previous concerns and questions, I recommend to publish this manuscript with minor corrections listed below:

1. line 126-127: station 2 is not a mainstream site according to the rest of the manuscript, please change.

Thank you. This was corrected, and we removed the word mainstem.

2. line 128: change "to Santarem" to "at Santarem"

Thank you. This was corrected.

3. line 772-772: Figure 2 (b) does not have have color-coded weight-average carbon number, please delete it from the caption.

Thank you. This was corrected.

4. line 897-900: Figure caption does not match the figure, please correct the caption.

Thank you. This was corrected.